# Wireless Measurements Using Electrical Impedance Spectroscopy to Monitor Fracture Healing

**DOI:** 10.3390/s22166233

**Published:** 2022-08-19

**Authors:** Naomasa Fukase, Victoria R. Duke, Monica C. Lin, Ingrid K. Stake, Matthieu Huard, Johnny Huard, Meir T. Marmor, Michel M. Maharbiz, Nicole P. Ehrhart, Chelsea S. Bahney, Safa T. Herfat

**Affiliations:** 1Linda and Mitch Hart Center for Regenerative & Personalized Medicine at the Steadman Philippon Research Institute, Vail, CO 81657, USA; 2UCSF Orthopaedic Trauma Institute, Zuckerberg San Francisco General Hospital, San Francisco, CA 94110, USA; 3Department of Bioengineering, University of California, Berkeley, CA 94720, USA; 4Department of Orthopaedic Surgery, Ostfold Hospital Trust, 1714 Graalum, Norway; 5Department of Electrical Engineering and Computer Sciences, University of California, Berkeley, CA 94720, USA; 6Chan Zuckerberg Biohub, San Francisco, CA 94158, USA; 7Department of Clinical Sciences, Flint Animal Cancer Center, College of Veterinary Medicine, Colorado State University, Fort Collins, CO 80523, USA

**Keywords:** bone fracture repair, electrical impedance spectroscopy, smart bone plate, wireless measurements, Bluetooth transmission, longitudinal monitoring

## Abstract

There is an unmet need for improved, clinically relevant methods to longitudinally quantify bone healing during fracture care. Here we develop a smart bone plate to wirelessly monitor healing utilizing electrical impedance spectroscopy (EIS) to provide real-time data on tissue composition within the fracture callus. To validate our technology, we created a 1-mm rabbit tibial defect and fixed the bone with a standard veterinary plate modified with a custom-designed housing that included two impedance sensors capable of wireless transmission. Impedance magnitude and phase measurements were transmitted every 48 h for up to 10 weeks. Bone healing was assessed by X-ray, µCT, and histology. Our results indicated the sensors successfully incorporated into the fracture callus and did not impede repair. Electrical impedance, resistance, and reactance increased steadily from weeks 3 to 7—corresponding to the transition from hematoma to cartilage to bone within the fracture gap—then plateaued as the bone began to consolidate. These three electrical readings significantly correlated with traditional measurements of bone healing and successfully distinguished between union and not-healed fractures, with the strongest relationship found with impedance magnitude. These results suggest that our EIS smart bone plate can provide continuous and highly sensitive quantitative tissue measurements throughout the course of fracture healing to better guide personalized clinical care.

## 1. Introduction

### 1.1. Fracture Healing Prevalence, Biology, and Economic Impact 

Musculoskeletal diseases incurred health care costs totaling $980 billion dollars in 2014 in the United States alone [1]. Given bone fractures are one of the most common injuries worldwide—with an estimated 189 million fractures occurring worldwide each year [2,3]—fracture healing presents a significant economic burden to the health care field. Complications in fracture healing such as delayed or nonunion develop in an estimated 10–15% of healthy individuals [4,5]. Incidence of poor healing is higher when the fracture pattern presents with large gaps, highly comminuted bones, lack of cortical continuity, or butterfly fragments [6,7]. Furthermore, patients with common comorbidities such as diabetes, obesity, smoking, malnutrition, osteoporosis, or being older than 65 can drive impaired healing rates towards 50% [8,9]. These prevalent and problematic fractures remain difficult and costly to treat [10,11,12].

Proper fracture healing is a complex and highly coordinated process where the mechanical fixation of the bone drives the molecular sequences of tissue differentiation [13]. In the small subset of bones that can achieve perfect reduction and rigid fixation, fractures are healed by intramembranous ossification where progenitor cells directly differentiate into bone [14]. However, in the vast majority of fractures, semi-rigid fixation is preferred and bone heals indirectly through a cartilage intermediate. This process of endochondral repair is traditionally defined by the transition to a native bone structure by proceeding through four distinct tissue types: (1) hematoma, (2) cartilage, (3) trabecular bone, (4) cortical bone [15,16]. The role of hematoma is to stop the bleeding, contain debris, and trigger a pro-inflammatory response that is critical to recruiting cells to the defect [17,18]. This fibrin-rich clot also forms a provisional scaffold for the proliferation and differentiation of cells to start the healing process. As inflammation resolves, the progenitor cells within the fracture gap differentiate into chondrocytes to generate a soft callus that serves functionally to provide temporary stability between the two bone ends. Chondrocytes in the fracture gap are different than those in the articular cartilage that line the ends of bone within the joint in that these chondrocytes undergo hypertrophic maturation to promote mineralization of the cartilage matrix. Hypertrophic chondrocytes also stimulate vascularization and innervation of the cartilage matrix, which initiates formation of trabecular bone through transformation of chondrocytes into osteoblasts [19]. Lastly, the newly formed trabecular bone remodels into cortical bone resembling that of the original long bone [20]. These defined stages of healing are well characterized histologically, however early stages of healing are not easily differentiable by standard, clinically relevant methods of monitoring fracture healing. 

A fracture becomes nonunion when the normal cascade of fracture healing is disrupted or fails [9]. The FDA’s standard definition of nonunion is a fracture that has lasted at least nine months with no signs of healing for three consecutive months [21]. Currently, the gold-standard in fracture care combines clinical examination and X-ray imaging to visualize tissue mineralization and evaluate healing progression. However, X-ray evaluation is subjective, problematic when implants obscure the fracture site, and most importantly, cannot provide diagnosis within the early stages of fracture healing [22]. This is particularly important as nonunion cases are suspected to biologically differ from standard healing cases prior to bone mineralization and remodeling. Since there are currently no methods to measure the biological state of healing, definitive diagnosis of nonunion often relies on computed tomography (CT) scans or magnetic resonance imaging (MRI) in the 6–9-month time frame. Not only are these modalities very expensive, but they are associated with significant radiation dosage, can be uncomfortable for the patient, and hardware artifacts often compromise their usefulness in diagnosis [23]. 

Treating patients with poorly healing fractures is a lengthy and costly process that requires physicians to assimilate qualitative data to develop a personalized treatment plan. If delayed healing is suspected, less invasive treatments such as ultrasound [24,25] or pulsed electromechanical field [26] may be applied prior to extensive revision surgery that carries risk of complications [27]. However, efficacy evidence for these non-operative procedures is limited and they appear to be time sensitive. Thus, a real-time, quantitative, non-invasive, and patient-specific method of monitoring fracture healing would significantly improve clinical fracture care, decrease the financial burden of extended nonunion intervention, and improve the formulation of patient-specific rehabilitation protocols. 

### 1.2. Advances in Fracture Monitoring Technology 

Novel approaches for monitoring fracture healing is an area of active research. Improving imaging modalities that enable quantitative assessment of healing is one approach under development [21,28]. Unfortunately, these systems only provide measurements at discrete timepoints and necessitate physicians or trained operators to conduct and analyze results. Furthermore, CT, positron emission tomography, and single-photon emission computed tomography and their combined contrast techniques, all expose patients to high levels of radiation. The development of diagnostic devices that could provide high resolution, injury-specific, and quantitative assessment of the fracture healing without frequent imaging would be clinically useful and prevent excessive radiation exposure. Additionally, integration within current hardware platforms would allow surgeons to continue standard of care treatment protocols, as hardware is not typically removed following surgery. While diagnostic hardware capabilities could add value for all patients suffering a fracture, this is largely a diagnostic modality targeting patients at a high risk for delayed or nonunion, including those with diabetes, a history of smoking or alcoholism, or an NSAID prescription. Increased data on healing status could also provide information related to responses to novel fracture treatments or possibly improve the understanding of when to administer time-sensitive biologics or when begin weight-bearing.

Biologic methods are also being developed to proactively diagnose nonunion fractures through the quantification of biomarkers specific to different phases of fracture healing. Clinical studies measuring alkaline phosphatase concentrations in the callus showed significant differences in union and nonunion cases [29]. However, this diagnosis timeline coincided with that provided by radiographs and therefore only serves as a complementary tool to X-rays rather than a supplementary one. Similarly, there are a number of bone turnover markers (BTMs) that have been explored for their utility in differentiating the rate of fracture repair [30]. Recently published preclinical [31] and clinical [32] studies suggest that certain BTMs may be potential surrogate markers of fracture healing, but additional validation is necessary to understand whether they can predict healing earlier than X-ray. Furthermore, bioassays require distinct blood collection and thus do not provide continuous feedback.

Mechanical assessment can also be a useful tool to leverage the increase in callus stiffness as the bone matures, thus providing earlier feedback on healing progression in the cartilage phase. Most favorable in this realm are implantable strain sensors that have the advantage of providing continuous feedback on callus stiffness throughout the healing timeline [1,33,34]. However, while the few proposed mechanical assessment systems compatible with internal fixation are promising, these designs only measure loads exerted on the implant and must infer the physiological load [35]; these inferences may not correspond to exact forces and moments acting in vivo [36]. Additionally, because each strain gauge only measures a single axis, the system requires several sensors, thus increasing the complexity and cost of the device to deliver a comprehensive overview of the fracture [37].

### 1.3. Electrical Impedance Spectroscopy (EIS) & Capacitive Sensors 

The response of biological tissues to electrical signals can be modeled as an impedance (Z), a frequency-dependent combination of resistance (R) and a reactance term (X). The reactance term in tissue is usually capacitive. This is expressed as a complex number denoted by the imaginary unit j:
Z = R + jX(1)

As a simplification, the primary contribution to the resistance term arises from charge motion through the ion-rich intra- and extracellular matrices, while the primary contribution to the (capacitive) reactance term arises from charge separation across lipid bilayer cell membranes and cell walls [38,39,40,41]. Building on previous work that utilized microscale sensors to demonstrate differences in electrical properties of blood, cartilage, and varyingly porous bone tissues in vitro, ex vivo, and in vivo [42,43], here we leverage EIS to characterize real-time progression of fracture repair in a rabbit model given the differences in electrical properties of the tissue composition throughout each stage of fracture healing.

While our previous study successfully implanted microscale, wired sensors for post-operative fracture monitoring in a mouse model [44], the small-scale sensors were prone to movement, required a connection to external wires to acquire measurements, and had low signal-to-noise ratios due to the small electrode size. Thus, for this study, we redesigned and optimized a fully wireless and watertight capacitive sensor system to integrate with commercially available bone plates. We then moved on to validate the utility of real-time wireless EIS measurements using a rabbit fracture model with a 1-mm segmental defect in the tibial shaft. EIS measurements were compared to standard X-ray, µCT, and histological measurements of fracture healing. The tibia was selected because it is at an elevated risk of impaired fracture healing, with nonunion rates between 10% to 18.5% in healthy patients [6,45,46]. Our study was designed to test the following hypotheses: (1) that this novel EIS platform could take continuous measurements over a clinically relevant time scale of more than 6 weeks, (2) that the sensors would not interfere with normal fracture healing, (3) that EIS measurements will increase with fracture healing, and (4) that EIS patterns will distinguish between good and delayed healing.

## 2. Materials and Methods

### 2.1. Sensor Design 

The sensor is a custom-made impedance sensing device equipped with a wireless radio module and two electrode wires. The module is made of two individual printed circuit boards (PCBs) that are connected through a mezzanine PCB connector (Figure 1B–D). Standard two-wire connection of the integrated circuit (AD5933) was used with an external resistor of 500 Ω. Specific components are listed below:

PCB 1 (38.5 × 7.5 mm): 

Complex impedance measurement integrated circuit (AD5933; Analog Devices, Wilmington, MA, USA) Analog multiplexer (ADG858; Analog Devices, Wilmington, MA, USA))Via holes for attachment of electrodes Electrode wires, 0.127 mm diameter Pt-Ir wire with 0.0381 mm insulation (#778000, A-M Systems, Sequim, WA, USA) 15 mAh single cell Lithium Polymer battery (GM300815; PowerStream, Orem, Utah, USA)

PCB 2 (28.4 × 7.5 mm): 

Radio module (CC1310; Texas Instruments, Dallas, TX, USA) Antenna, 915 MHz (66089-0906; Anaren, East Syracuse, NY, USA) Load switch (TPS27081A; Texas Instruments, Dallas, TX, USA)Voltage regulator (TPS78236; Texas Instruments, Dallas, TX, USA)

An off-the-shelf, miniature lithium-ion battery was soldered onto PCB 1. Electrode wires (9 mm and 14 mm in length) were attached to the via holes on PCB 2 using silver epoxy (AB2902 Angstrombond; Angstrom Sciences, Inc., Duquesne, PA, USA). Prior to attaching the two boards for final potting, the batteries were fully charged. Custom two-part sensor housings were designed and additively manufactured by a 3D printer (Form 3b; FormLabs, Somerville, MA, USA) using biocompatible resin (BioMed Clear; Formlabs, Somerville, MA, USA). The two boards were attached and inserted into the bottom housing component. Biocompatible resin (301; EPO-TEK) was poured into one side of the housing to encapsulate the electronics (Figure 1E,F). Insulation was only removed from the ends of the two electrode wires after curing, with a final exposed surface area of approximately 0.67 mm^2^ per wire. The other side of the housing was placed onto the cured bottom assembly and packed for shipping and sterilization (Figure 1G). The sensors were sterilized using a low temperature sterilization system (STERRAD; Advanced Sterilization Products) prior to implantation.

### 2.2. Experimental Fracture Fixation—Surgical Planning 

Prior to the in vivo trials, an off-the-shelf, nine-hole stainless steel locking plate (Synthes VP4012.09; 2.0 mm LCP plate) was placed on a cadaveric rabbit tibia sample to determine the optimal location for bone plate fixation. The medial surface of the tibia was determined to have a sufficiently flat surface, which was optimal for plate placement (Figure 2A,B). It is important to note that while this study utilized a Synthes locking plate, the sensor is designed to be compatible with any standard surgical plate using any surgical screws at the physician’s discretion. 

### 2.3. Animal Model—Rabbit 

Six mature New Zealand White Rabbits (all female, weight of approximately 3 kg) were used for the in vivo study. Animals were housed in a controlled temperature (65–71 °F) at a 12/12-h light/dark cycle with food and water available ad libitum. Surgical procedures were performed by an experienced orthopedic surgeon and followed Institutional Animal Care and Use Committee approved protocols (IACUC #1337). 

### 2.4. Experimental Fracture Fixation 

Under general anesthesia, a skin incision was made on the lateral side of the right tibia. A nine-hole stainless steel locking bone plate (Synthes VP4012.09; 2.0 mm LCP plate) was then temporarily placed on the medial side of the tibia and fixed with three proximal and three distal screws. Cortical bone screws (Synthes VS202; 2.0 mm diameter, 12/14 mm lengths) were inserted into the two proximal holes and the two distal holes and locking screws (Synthes VS207; 2.0 mm diameter, 12 mm length) were inserted into the third hole from both the proximal distal ends. The central holes remained undrilled at this stage. The screw lengths were determined using a depth gauge as is standard clinical procedure. Next, the osteotomy position was accurately determined by marking the medial cortex of the tibia with a microdrill at the center of the fifth hole from the proximal side (i.e., center of the bone plate). The plate was then removed, and an osteotomy was created in the tibia at the position of the landmark. A 1-mm segmental defect was created using an oscillating bone saw with saline irrigation to reduce risk of thermal necrosis. The locking plate was then placed on the medial side of the tibia to coincide with the previously drilled holes and fixed with two locking screws (third hole from the proximal and distal ends) to ensure that alignment and a 1-mm bone defect was maintained at the osteotomy site. The sensor housing was then attached to the bone plate (Figure 2C) using four cortical bone screws (two proximal and two distal). After thorough rinsing with saline, the two electrode wires were inserted precisely into the bone defect through the fifth hole in the housing/plate and positioned inside the bone gap (Figure 2D). Finally, the sensor was press-fitted into the housing to complete the assembly. 

Throughout the procedure, the tibialis anterior muscle was carefully preserved to maintain foot function. However, during the first surgery we observed necrosis of the skin as well as compartment syndrome. To minimize necrosis, promote wound healing, and prevent infection, a lateral approach was subsequently used to ensure the skin incision was not directly above the implant. Additionally, to prevent compartment syndrome, soft tissue coverage of the implant was kept to a minimum. Since the implant cannot be adequately covered with soft tissue, care was taken to avoid post-operative infection. Following the operation, a suture with non-absorbable thread was utilized to close the soft tissue and skin. Although the thickness and poor elasticity of rabbit skin of the unit resulted in some strain during suturing, the unit was completely covered with soft tissue and the wound was successfully closed (Figure 2E). 

### 2.5. EIS Data Acquisition and Processing

Impedance magnitude (|Z|) and phase (θ) measurements were recorded every 48 h until the rabbits were sacrificed. The AD5933 was turned off between measurements to save power, then reconfigured by the microcontroller embedded in the radio module on every wake-up cycle, which performed measurements, collected data, and wirelessly transmitted results to a laptop. All impedance (Z) values being reported are impedance magnitudes:(2)$$|Z|=\sqrt{R^{2}+X^{2}}$$

Resistance and reactance were calculated as follows: R = |Z|cos(θ)(3)
X = |Z|sin(θ)(4)

Two-point impedance measurements were taken using a 1.98 V peak-to-peak sine wave output signal, and a frequency sweep between 2 and 99 kHz was utilized for the initial measurements to determine the optimum frequency. Discrepancies between varying tissue compositions were largest at low frequencies, thus all subsequent recordings were collected at 2 kHz. At this frequency, current ranged between 0.08–1.01 mA, which is well below the 10 mA safety limit for electrical measures in humans. Variations in temperature or voltage were controlled for by using the multiplexer (ADG858) to calibrate the device prior to each series of measurements. 

For the purpose of comparison between samples, all electrical measures were normalized as a ratio to the common initial measurement day, 13 days post-operative, to allow the injury to stabilize.

### 2.6. Medical Imaging 

Radiographs of the anteroposterior (AP), lateral, and oblique views were taken under sedation on the first post-operative day, followed by biweekly images thereafter until euthanasia to monitor the fracture healing process and confirm that both electrode wires remained within the fracture gap. After euthanasia, morphological appearance and morphometric analysis were performed by ex vivo µCT using a Scanco VivaCT 80 µCT scanner (Scanco Medical AG, CH-8306 Bruettiselllen) with the following settings: 70 kVp, 0.5 mm AL, BH: 1200 mg HA/cm^3^, scaling 4096; voxel size 15.6 µm. Images were reconstructed using Scanco Medical Software. Data analysis including bone volume (BV), bone volume density (BV/TV), and the trabecular number (Tb.N) was performed. 

### 2.7. Three Cortices Radiographic Scoring 

AP, lateral, and oblique radiographs were blindly evaluated by three qualified orthopedic surgeons at each timepoint. The three cortices were scored as (1) no callus, (2) callus present, (3) bridging callus, and (4) remodeled. Scores were then summed and averaged across the three physicians with final, customized three cortices radiographic scores ranging between 3 and 12. Tibias were defined as union when radiograph score ≥ 10, a ratio that is in accordance with 90% of clinicians who define the standard mRUST of ≥13/16 to be united [47]. 

### 2.8. Histology

The tibiae of post-medical-imaging rabbits were surgically removed and degloved 6-, 8-, and 10-weeks post-fracture. At time of collection, tibias were fixed in 4% paraformaldehyde and decalcified for 28 days in 19% ethylenediaminetetraacetic acid (EDTA) with rocking. Excess bone beyond the fracture region was removed, and the regions of interest were further decalcified for 14 days in 19% EDTA followed by 7 days in 5% formic acid. All solutions were changed every other day and stored at 4 °C throughout the decalcification process. Tibias were processed for paraffin embedding, serial sectioned into 8 µm sections, and stained with Hall-Brundt’s Quadruple (HBQ) to visualize bone (red), cartilage (blue), and fibrous tissue (purple) according to standard protocol. Sections were mounted with Permount^TM^ mounting medium and sealed with coverslips. Representative brightfield images of 6-, 8-, and 10-weeks post-op were captured at 2X (stitched) and 4X magnification on a Nikon Eclipse Ni-U microscope with Nikon NIS Basic Research Elements Software version 4.30 (Nikon Instruments Inc., Melville, NY, USA).

### 2.9. Statistical Analysis 

Normalized measurements were fit to a sigmoidal (four parameter logistic) curve to approximate the standard fracture healing curve [48,49,50]. Data were tested for normality and significance testing was performed with either paired, Wilcoxon rank sum testing (α = 0.05, significance *p* < 0.05) or one-way ANOVA followed by Tukey post-hoc testing to adjust for multiple comparisons. Spearman rank-based correlation coefficients were calculated between radiograph scores and normalized sensor measurements given the nonlinear relationship. All statistical modelling and testing were performed using Prism software version 9.3.1 (GraphPad Software, San Diego, CA, USA). 

## 3. Results

### 3.1. Smart Fracture Fixation Implants Do Not Impede Standard Fracture Healing 

For this study we followed a previously established, reproducible experimental technique for creating fractures in the mid-diaphysis of the rabbit tibia [51]. Fractures were secured using a standard nine-hole stainless steel veterinary locking plate fit with our custom-built electrical impedance spectroscopy (EIS) sensor system (Figure 1 and Figure 2). Electrode wires were inserted into the fracture gap and the surgeon confirmed by radiograph that 12/12 wires—two per sensor—were correctly positioned immediately post-operatively. Fracture healing was semi-quantitated by three blinded orthopaedic surgeons from longitudinal radiographs using a standard four-point scoring system, defined in the methods, over three bone cortices. Union was defined as an average radiographic score ≥ 10; by this definition union was observed in 0/6 cases at 4 weeks, 1/6 cases at 6 weeks, 3/4 cases at 8 weeks, and 2/3 cases at 10 weeks post-op (Figure 3). Longitudinal wire placement was also followed by radiograph. We found only a single case in which one of the two wires deviated from the gap at 2 weeks post-op, however, the other wire was confirmed to remain in the gap until the rabbit was euthanized at 6 weeks post-op.

µCT was performed as an additional measure of fracture healing to obtain quantitative data related to bone mineral density (BMD) and bone volume/total volume (BV/TV). Data parallel radiographic scoring (Figure 3B,C) with a steady increase in quantitative metrics of bone repair throughout the time course of healing (Figure 4A,B). Representative µCT and histology sections visually confirm the progression of trabeculated new bone to remodeled cortical bone within the fracture gap (Figure 4C–H). There was no visual disruption to healing progression in the areas around the sensor. 

### 3.2. Feasibility of Wireless Readings 

In addition to confirming that the electrode wires did not interfere with healing progression, our study aimed to understand the capabilities and lifespan of our wireless sensors. First, we performed an ex vivo experiment to create the fracture model, implant the fracture fixation hardware, and confirm that wireless reading transmitted through the skin. We confirmed consistent, strong sensor readings to a computer system up to 3 m in distance from the implanted bone.

For in vivo application, the battery was connected to the PCB in the sensor system, then the device was sterilized and surgically implanted within 48 h to preserve the power source. The STERRAD sterilized EIS sensor system was shipped to the surgical facility in a custom-designed 3D-printed packaging container (Figure 1G). With a full charge, the sensor was designed to acquire one wireless measurement every 24 h for a maximum of 8 weeks. For this study we chose to record measurements every 48 h with the aim of extending the sensor’s life beyond 8 weeks. For 2/6 cases (rabbits U4879 and U4881), both reached union and provided complete sensor data for the entirety of the experiment (Figure 5). In 3/6 cases the sensors yielded data for only 50–75% of the experiment, with measurement extending to between 3 and 4.5 weeks. Also, 1/6 of the sensors (U4880) did not provide any data following implantation. This sensor was confirmed to work prior to implantation, so it is presumed that the electrode wires were compromised during the surgery either within the fracture gap or at the connection to the PCB. While ex vivo measurements could be reliably collected at up to 3 m, we found that some daily measurements were lost throughout the longitudinal study due to interference from the metal on the animal housing cages. Interference was mitigated by moving the computer to within 1 m of the sensors and placing the computer in a position that would reduce metal interference. 

### 3.3. Wireless Sensor Measurements Capture Fracture Healing Progression 

In vivo longitudinal wireless sensor readings were performed across a frequency sweep from 2 kHz to 99 kHz to obtain impedance magnitude (Z), resistance (R), and reactance (X). The difference in the measurement was calculated as the final measurement minus initial measurement (for example: Z_final_–Z_initial_) from all five functioning sensors. This differential shows that the lowest recording frequency of 2 kHz provided the greatest spread in the electrical measures and therefore better sensitivity to distinguish between different tissue types (Figure 5A–C). Based on this finding, subsequent data are only presented for the 2 kHz recording frequency unless otherwise noted.

On the initial measurement day using 2 kHz, impedance magnitudes ranged from 2.6 to 4.2 kΩ, resistance from 1.7 to 3.7 kΩ, and reactance from −2.0 to −1.0 kΩ. Over 8 weeks of healing, the ranges of raw impedance, resistance, and reactance measurements were 15.9 ± 8.0 kΩ, 14.8 ± 7.4 kΩ, and 8.3 ± 4.8 kΩ, respectively (mean ± standard deviation). For the purposes of comparison across samples, electrical measures were normalized (“norm”) as a ratio to the common initial measurement day. While it is possible that the distance between wires could change during the initial days post-op when the fracture area is filled with fluid, it is expected that the wires should be held in place once a hematoma forms. By normalizing all sensor measurements to day 13, the inter-electrode distance is expected to be stable for the remainder of the measurement period.

Normalized data were fit to a sigmoid function to model the standard fracture healing curve as previously recommended [48,50]. Akin to our prior in vivo murine studies [44], impedance and resistance values for the two fractures with complete sensor measurements grew sigmoidally over time until union was achieved (Figure 5D,E). After an initially slow increase in electrical measures for about 20 days, the curves deflect and the rate at which electrical measurements increase accelerates. Peak impedance occurred 57 and 55 days post-fracture for samples U4881 and U4879, respectively, then plateaued. Resistance and impedance approach statistically similar upper limits in the healed bone tissue across both of these sensors. Similar to resistance and impedance, reactance measurements grow steadily over time but present with a less distinctive sigmoidal curve with healing (Figure 5F).

Electrical measurements’ patterns for resistance and impedance were maintained across all frequencies (Figure 5G,H). However, the goodness of fit (R^2^) to a sigmoidal curve decreased with increasing frequency (Figure 5I), further supporting the use of 2 kHz frequency.

As we have done previously [44], the biological tissue being measured can be described as an equivalent circuit below, modeling the ionic intracellular environment as a resistor and the double-layer cell membrane as a capacitor. These two elements are first placed in parallel, and then in series with another resistor that reflects the ionic extracellular environment. A constant phase element can be added in series to model the electrode–electrolyte interface. In this paper, we find that the overall impedance of this system accurately delineates healing and non-healing fractures.



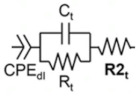



### 3.4. Sensor Measurements Monitor Real-Time Fracture Healing & Distinguish between Union and Not-Healed Fractures 

In all samples, radiographic scores were found to significantly correlate to the three electrical measurements, with the strongest relationship with impedance (Figure 6A–C,G). Additionally, in the three sensors that produced measurements at the time of µCT, all sensor measurements positively trended with BMD and to a lesser extent BV/TV (Figure 6D,E). Trabecular number and each sensor measurement had strong, negative relationships (Figure 6F); no statistical difference was evident with any µCT outcome due to the limited sample size (Figure 6G). 

Both impedance and resistance measurements readily distinguish between the cases that were categorized as not healed (radiographic score < 10) and union (radiographic score ≥ 10) (Figure 7A,B). Statistical difference between not healed and union fracture was less clear with reactance, paralleling the shallower temporal reactance measurements (Figure 5F and Figure 7C,D).

## 4. Discussion

This study was designed to address the unmet clinical need for quantitative, longitudinal assessments of the fracture callus for the purpose of monitoring healing progression. To the best of our knowledge, this is the first study to design and fabricate a microscale sensor implanted directly into the fracture gap intraoperatively for wireless, highly sensitive, repeated, in vivo electrical impedance spectroscopy (EIS) measurements of fracture healing. Importantly, our results demonstrate that EIS measurements correlate with bone healing progression and that the microscale electrode wires integrate within the fracture gap without interfering with the endogenous repair process. 

In this study we scaled up from our previous murine work [44] to use a rabbit model of fracture repair that allowed us to interface our custom-built EIS sensor system with commercially available bone plates and electronic components that enable wireless data collection. With this appropriately scaled system, we confirmed the findings in our earlier studies showing that electrical properties of the fracture callus temporally change with tissue composition [42,44]. Specifically, we saw that fracture union is associated with sigmoidal temporal response across resistance, reactance, and impedance measurements that approach a horizontal asymptote with fracture union. 

Conceptually, resistance (R) reflects the ability of a material to carry an electrical charge, with highly conductive materials having low resistance and less conductive materials having high resistance. Therefore, our finding that resistance increased with the radiographic formation of bone reflects changes in the tissue-specific properties as the highly conductive blood remodels into a dense bone structure. We observed less change in the resistance during the bone remodeling phase of fracture repair. Compared to trabecular bone, cortical bone has lower fluid content and less surface area exposed to bone marrow and blood flow [52], which decreases conductivity and increases resistance. However, cortical bone is also associated with a higher calcium content [53,54]—an extremely conductive ion—that may counteract the loss of the bone marrow, resulting in normalization of electrical resistance with bone remodeling. Since the bone remodeling phase of repair is easily detectable through radiographic imaging, the static nature of the electrical readings at this point is not problematic and further supports that the clinical utility of EIS is during the early phases of fracture repair. 

Reactance (X) represents the effect of storing energy, including charge separation across the cell membrane (which acts as a barrier to the flow of charge and appears capacitive). Tissues with a higher number of cells have a greater ability to store energy, resulting in a greater reactance, an effect that is amplified at low frequencies. Since bone consists of a capacitive matrix layer and bone marrow is highly cellular, it is reasonable to assume that reactance increased preferentially in well-healed tissues where there is good cellular connectivity across the fracture gap following bridging. As the callus remodeled from porous trabecular bone into more densely cellular cortical (compact) bone [53], reactance measurements continued to increase and did not reach the distinctive plateau observed with resistance. Furthermore, the magnitude of the difference between the starting and ending point of the sigmoidal curve was less significant with reactance than for resistance. Taken together, this suggests that these cellular-derived electrical changes drive less electrical response than matrix conductivity.

Accordingly, we also see that impedance (Z)—the combination of both resistance (R) and reactance (X)—increases during the time course of fracture healing and may provide real-time information on the repair progression. Our data further suggest that the slope in the linear phase of the sigmoidal electrical response curve is indicative of whether bone healing is well, versus slowly progressing (low slope, delayed union) or completely non-progressing (nonunion) (Figure 8). Throughout the study we saw the strongest statistical correlations between impendence and healing, relative to resistance and reactance, indicating this combined matrix and cellular measurement is most useful.

These electrical response patterns are consistent with our previous study testing wired sensor implants in a murine fracture model [44]. The major challenge in the mouse fracture healing model was designing a smart bone plate within the small size constraint of that model. Moreover, we experienced model-specific difficulties as the mice chewed and damaged the sensors, making data collection during the healing period difficult and resulting in the removal of data from animals with broken sensors. Even within the mouse model our microscale sensors enabled integration into the fracture tissue, however, increased movement and higher contact impedance of the electrodes could reduce the signal-to-noise ratio of the measurements. Therefore, to mitigate this effect, we employed stiffer electrodes, increased the surface area of the electrodes, and integrated the custom-built smart wireless sensors into a commercially available veterinary bone plate using a larger animal, the New Zealand White Rabbit. The use of stiffer electrodes minimized the potential for electrode movement within the fracture gap, thus maintaining inter-electrode distance, decreasing scarring, and promoting more consistent sensor measurements of the tissue. Despite the increase in total electrode surface area, the wires appeared to be sufficiently thin to not induce scarring or fibrous tissue, as confirmed by post-operative histological images and X-ray. Furthermore, because the injury is stabilized during the early stages of healing, the inter-electrode distance is anticipated to remain constant throughout the course of healing. Given that the wires did not impair tissue repair, mechanical modifications could be envisioned that could further secure the electrode distance within the fracture callus. Consistent impedance measurements during the early parts of healing in this experiment as well as previously measured baseline impedance in saline solution [44] both support that any change in cell constant caused by relative movement of the electrodes can be presumed negligible. Taken together these improvements to the sensor technology facilitate wireless transfer of real-time EIS data, bringing the system closer to clinical application.

While the rabbit model enabled proof-of-concept for a fully translational smart bone plate design, we still experienced some technical difficulties that presented as limitations of the studies. First were challenges associated with the animal model itself. The rabbits’ surgical site was prone to infection and their thin skin did not stretch well over the housing. These were addressed with the improved surgical approach as detailed in the methods, but did lead to a more limited sample size due to humane euthanasia requirements for some of our animals. We also did not have 100% reliability of our daily wireless data transmission and three of six sensors only produced data for ~1 month. We did determine that wireless data transmission was sporadically disrupted due to interference with the metal wire cages, but this did not account for permanent disruption of some of the sensors, which may have been due to damaged sensors. Increasing the sensor thickness, electrode wire diameter, or battery capacity might be feasible in this larger animal model or at a human-scale to reduce the likelihood of sensor damage and increase data reliability. Alternatively, changes to the data acquisition program might also increase sensor life as daily readings are not necessary to obtain clinically relevant resolution on healing patterns. Additional scaling of the current technology to adapt to human scale is not anticipated to be challenging as the increased footprint on the sensor board will only make the design easier.

We believe this sensor technology could have a significant clinical impact to improve patient care. Several advanced imaging techniques have been explored to gain more information about the biology of bone healing. For example, bone scintigraphy (BS)—a nuclear medicine imaging technique—uses uptake of BS radiotracers to assess bone metabolism and turnover, enabling diagnosis and monitoring of bone diseases, infection, and fractures [55,56,57,58,59]. Despite the advantages of BS in assessing biological activity, a quantitative assessment of biological activity with scintigraphy is difficult to achieve [60]. Recently, Bhure et al. reported on the value of single-photon emission computed tomography/computed tomography (SPECT/CT) in the evaluation of necrotic bone fragments in patients with delayed bone healing or nonunion following traumatic fractures and concluded that SPECT/CT is an accurate tool in the evaluation of such patients and is superior to conventional planar BS [61]. In addition, Oe et al. found that standardized uptake value (SUV) can be quantified from SPECT of nonunion sites, indicating that quantitative SPECT parameters including SUVmax, SUVpeak, and SUVmean provide a potentially useful tool for assessing the metabolic activity of nonunion [62]. Nevertheless, these studies have mainly examined the need for autologous bone grafting to allow for appropriate surgical intervention, leaving unresolved the issue of distinguishing between delayed union and nonunion. Additionally, BS and SPECT/CT are not used clinically and are unlikely to be clinically adopted due to the increase in radiation to the patient.

The results of the current study provide a significant step forward in the development of wireless smart implants for monitoring the early progression of bone healing in patients. With greater sensitivity to the early stages of hematoma and cartilage than conventional techniques, this device could be particularly useful in detecting the early biological activity that distinguishes these two types of fractures, allowing physicians to be better informed about treatment options at earlier timepoints in the fracture healing process. Pseudo-real time EIS data could be wirelessly transmitted to a cell phone application, providing physicians updates on the status of fracture healing in line with modern glucose and electrocardiogram monitoring devices. While real-time EIS data are possible, it is not clinically necessary as the timescale of fracture healing is typically on the order of 6–12 weeks with standard clinical visits ranging from every 2–4 weeks; as such, daily or even weekly measurements would significantly enhance current resolution of healing.

Although additional research is needed to optimize the system prior to clinical implimentation, the miniaturization of the device facilitates its integration into conventional orthopedic implants, and our impedance measurements allow for highly sensitive tracking of these changes, providing sufficient information for clinical use without the extra expense or radiation exposure. Translating the EIS data to a clinically familiar value, such as a radiographic score, could further improve the clinical utility of the smart bone plate. While we provide very early correlative relationships in this study, more comprehensive clinical data would increase the reliability and robustness of future models.

## 5. Conclusions

The primary goal of this study was to develop and validate the technology for a translationally relevant smart fracture fixation device that uses EIS to provide quantitative information about the status of fracture repair in individuals. 

In this study, a fully implantable sensor technology, segmental defect fracture model, and fracture fixation strategy were developed to measure the electrical patterns throughout the early progression of fracture healing. The outcomes of this study are key steps forward in the development of a wireless smart bone plate for monitoring the early progression of bone healing in patients. These electrical measurements could provide clinicians with early quantitative information about the state of fracture repair and empower them to intervene earlier if there is evidence of compromised healing.

## 6. Patents

This work has a patent granted, WO2017030900A1 published 23 February 2017. It is available at https://patents.google.com/patent/WO2017030900A1. 

## Figures and Tables

**Figure 1 sensors-22-06233-f001:**
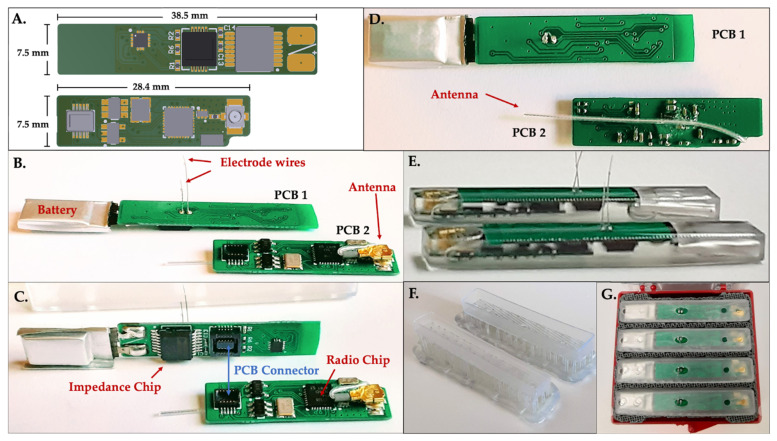
Smart sensor design and accompanying components. (**A**) CAD rendering of component PCBs with dimensions. (**B**–**D**) Component PCBs and (**E**) assembled sensors encapsulated in resin housing. (**F**,**G**) Custom 3D printed packaging.

**Figure 2 sensors-22-06233-f002:**
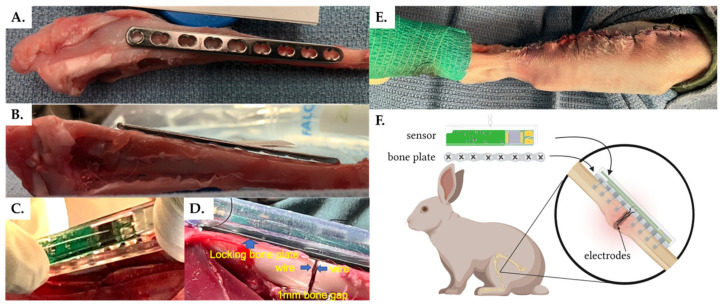
Experimental sensor implantation. (**A**,**B**) Surgical planning and sensor integration with (**C**,**D**) the bone fixation plate and (**E**) within the rabbit fracture model. (**F***) System overview. * Created with BioRender.com.

**Figure 3 sensors-22-06233-f003:**
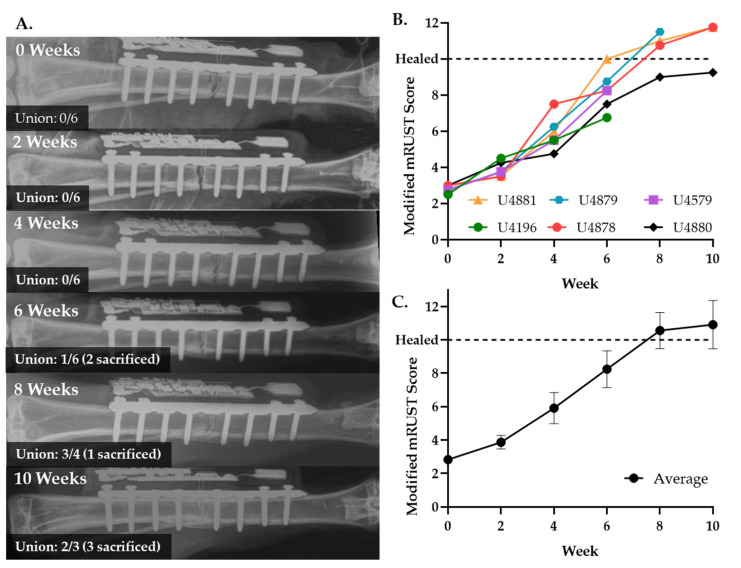
Rabbit tibiae healing by ~8 weeks with EIS smart bone plate. (**A**) Representative X-rays of healing weeks 2–10 weeks post-operatively. (**B**) Radiographic healing scores from individual animals. (**C**) Average radiographic healing scores over time.

**Figure 4 sensors-22-06233-f004:**
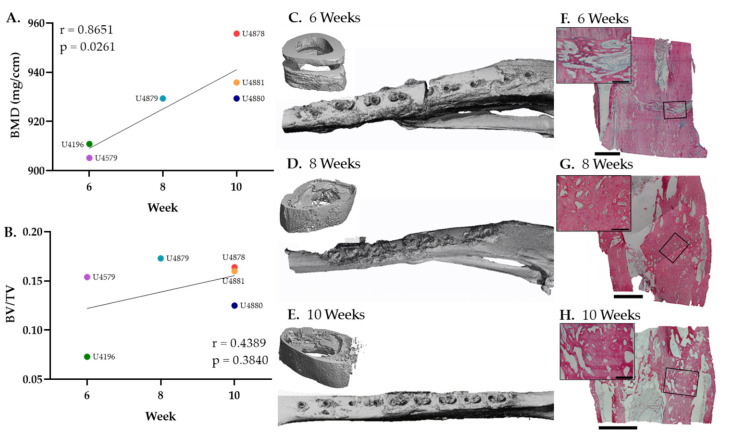
Quantitative bone mineral density (BMD) and bone volume (BV) consistently increase during bone repair. (**A**) BMD and (**B**) BV/TV measurements of individual animals at necropsy with corresponding correlation coefficients (r) and *p*-values (*p*). (**C**–**E**) Representative µCT images at 6-, 8-, and 10-week necropsy. (**F**–**H***) Representative HBQ stained histology images at 6-, 8-, and 10-week necropsy. * Scale bars are 500 (inset) and 2500 µm (bottom).

**Figure 5 sensors-22-06233-f005:**
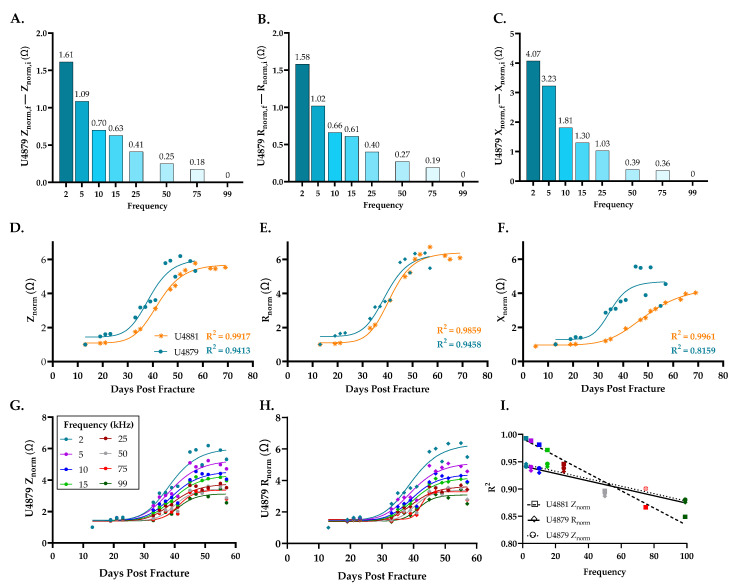
Normal fracture healing produces sigmoidal electrical responses at 2 kHz. (**A**–**C**) Difference in electrical measurements between final and initial recordings across all frequencies for sensor U4879 (blue). (**D**–**F**) Longitudinal electrical output at 2 kHz for sensor U4879 (blue) and U4881 (orange). (**G**,**H**) Electrical measurements across 2–99 kHz frequency sweep (colors indicated in ledged) for sensor U4879. (**I**) Goodness of fit (R^2^) to a sigmoidal curve for sensors across 2–99 kHz frequency sweep (colors indicated in ledged in G).

**Figure 6 sensors-22-06233-f006:**
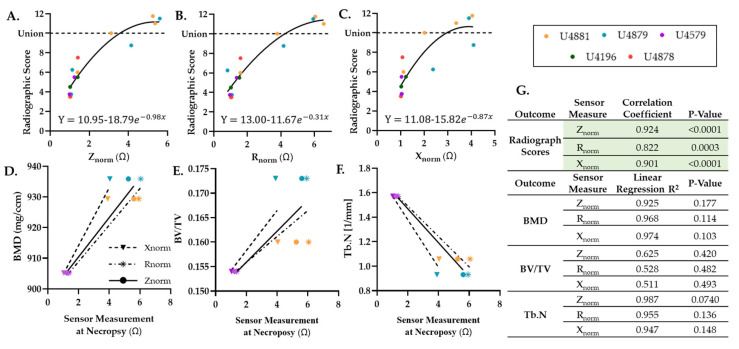
Smart sensors reproduce classic X-ray and µCT measures. All sensor measurements significantly correlate to (**A**–**C**) radiographic and (**D**–**F**) µCT outcomes. (**G**) Spearman correlation coefficients and goodness of fit to a linear regression (R^2^) between each measurement. Green coloring highlights statistical significance of P < 0.05.

**Figure 7 sensors-22-06233-f007:**
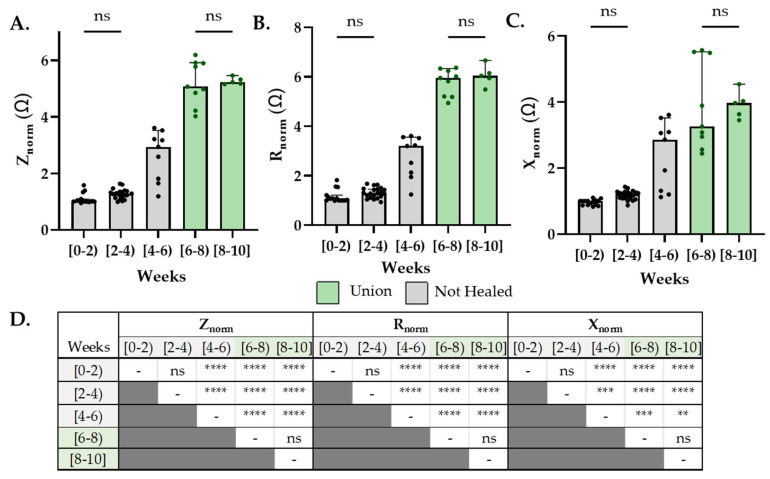
Sensors distinguish between union and not-healed fractures at all healing timepoints. (**A**–**C**) Sensor measurements grouped by healing timeframe (grey = not healed; green = union). (**D**) Only ns (no significance) shown on graph, all other comparisons are significant (α = 0.05). Internal references are greyed out on graph, green highlight indicates healed fractures. Data analyzed by one-way ANOVA followed by Tukey post-hoc testing. ** = P < 0.01, *** = P < 0.005, **** = P < 0.001.

**Figure 8 sensors-22-06233-f008:**
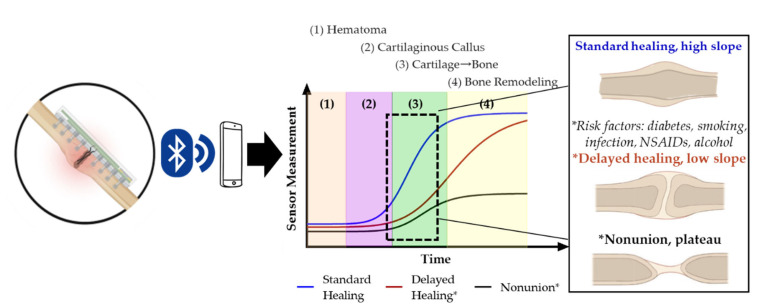
Pictorial representation of theoretical standard, delayed, and nonunion healing curves [9].

## Data Availability

The data presented in this study are available upon request from the corresponding authors.

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
