# Peer review of "Wireless Measurements Using Electrical Impedance Spectroscopy to Monitor Fracture Healing"

_sensors, 2022, doi:10.3390/s22166233_

Round 1
Reviewer 1 Report
The paper describes a device, an experimental protocol and the validating results that demonstrate the feasibility to wirelessly monitor healing of bone fractures utilizing real-time electrical impedance spectroscopy (EIS). The paper is well written and the methods appear to be rigorous and the results original.
There is however, in my opinion, an imbalance between the degree of description of the measurement protocols and the reference measurement methods, on one side, and the EIS measurement apparatus and methods, on the other side. Being the novelty of the work centered on the EIS methods and results, this aspect should be corrected. Specifically:
- In one of the pictures of figure 1, please include the size of the boards
- I assume that the standard 2-wire connection of the AD5933 is used. In this case, an external resistor which converts current to voltage (or which builds a half-bridge with the impedance under measurement) is used. Please describe if this is the setup and state the value of this resistor if it is the case. Other authors use an external front-end with the AD5933.
- Apparently, from the system description, each implanted device measures only through a pair of electrodes. Which is the role of the ADG858 multiplexer?
- The management of the measures (timing, configuration of AD5933 registers, collecting and sending the measurements) is performed by the microcontroller of the radio module or is managed by the external reading system and the AD5933 is configured every time it has to take a measurement? Is the power of the AD5933 switched off between measurements?
- The insulation of the wires between the measurement system and the fracture location is only provided by the application of the biocompatible resin? Can you provide an estimation of the insulation thickness? Can the distance between wires (in the path between the device and the measurement location) change along an experiment?
- In page 6, lines 259-263 the authors state that a voltage of 1,98 Vpp is used to perform the measurements. Can you provide an estimation of the resulting current? The electrical safety limitations in bioimpedance measurements are given in current. A rough conversion can be done just dividing the applied voltage by the maximum and minimum obtained impedances, which are not described in the paper. Being a 2-wire measurement, they would include both the tissue and the electrode impedances. It would be good to know if the current is close to the one allowed in measurements in humans according to IEC 60601-1:2005, in order to infer that a similar SNR would be obtained.
- If the batteries are connected after sterilization and transport (page 9), how is the battery contact area insulated and sterilized again?
- The results are normalized by subtraction of the initial value, which is a reasonable approach. The relative variation would also be useful. The information about the measured impedance and its dispersion is, however, missed. It would be good to provide just a text informing about the base impedance value ranges and its dispersion among cases, before normalization. This would provide information to the reader about how big is this dispersion compared with the changes obtained along the healing process.
- The authors have found that the biggest changes and correlations are obtained at the lowest frequency. This is reasonable in very controlled experimental conditions but the low frequency (2 kHz) is also where the electrode impedance is bigger (into the measurement frequency range) and can change the most. Using a 2-electrode schema, the two electrode impedances are included in the measurement. It would be really useful to have an estimation of the electrode impedance value. It can be easily measured using a saline solution or an agar block with a conductivity close to the one of the tissue under measurement and using a counter-electrode a lot bigger than the wire tip used as electrode. This will allow the reader to assess the relevance of the observed changes along the healing process. Also to reduce the reasonable doubt about if the changes are produced in the tissue or in the electrode impedances. Please include the measurements and a discussion about them.
- Another possible factor that would affect the measurements is the change in the cell constant. The distance between the electrodes would be different among experiments and can change along the process and the thickness of the scar area can also change along the process. I acknowledge the difficulty of performing that kind of measurements (I have performed similar ones in other organs). Can you please develop a bit more the discussion about this? The results are consistent but a measurement based on a single frequency is not robust against changes in the cell constant. Can the X-ray images or the post euthanasia analysis verify this distance?
- A side comment: I personally find more interesting the result in the reactance, which is more robust to the cell constant changes, as it would probably be the relative variation among frequencies or the phase angle. The behavior of X or phase angle would be probably more relevant at a mid-frequency, like 40 or 50 kHz.
Other aspects to be considered:
- The first sentence of the Introduction: “Musculoskeletal diseases incur more health care costs than breast, lung, and prostate 37 cancer combined, totaling $980 billion dollars in 2014 [1]” is, in my opinion, a bit inappropriate. The importance of the costs of the musculoskeletal diseases can be highlighted without comparing it with the cancer diseases.
- In the introduction, page 3, lines 101-105, the authors claim that “…implantable device precisely at the location of injury that allows for immediate quantitative assessment of the fracture healing process would enable clinicians to better diagnose healing progression and treat the delay in bone healing even more rapidly than conventional methods. Reducing the number of unnecessary radiological examinations would also contribute to minimizing health care costs.” This suggests that in a mid-long term, an evolved version of the implantable device can be routinely used to monitor bone fracture healing in humans. In my opinion, it would be good to add some reflections on the implications of this possibility. Would it be intended for research purposes? To assess the effectivity of new treatments? For every fracture? Would this mean that a new operation would be needed to extract the device after healing?
- In page 3, line 144, the authors use two self-references to support the very basic concepts of bioimpedance. In my opinion, to state the dependence of electrical impedance on the basic conduction mechanisms across intra and extra-cellular milieus and cell membranes, some of the very basic references should be used (i.e. works of prof. Schwann, Gabriel, Gersing, or reference books as Grimnes & Martinssen).
Reviewer 2 Report
Wirelessly monitoring is important for medical research. I think it’s interesting and important research. I would recommend it if these issues are handled properly.
1. Could your devices acquire real-time EIS information? Please provide relevant indicators in detail, such as latency time.
2. How do you process your EIS results? Please provide equivalent circuits and explain that.
3. I suggest the author build a semi-quantitative relationship between X-rays images and EIS results.
4. Highlight the novelties of this manuscript since you already have a patent granted.
Round 2
Reviewer 1 Report
In my opinion, the authors have put a lot of effort into carefully analyzing and answering my previous observations in a short time. This can be only done if there exists a solid background and a lot of additional materials that support the ones published in the paper. Not all my observations have been clarified whit the same deepness but I acknowledge the difficulty of doing so and, at least, the authors have provided a reasonable explanation to all my questions. The results reported in the paper are very consistent and I agree with the authors in assuming that most of the possible sources of uncertainty are small if not negligible.
In my opinion, the paper can be published as is. Congratulations to the authors.